# Dose Tapering Strategy for Heroin Abstinence among Methadone Maintenance Treatment Participants: Evidence from A Retrospective Study in Guangdong, China

**DOI:** 10.3390/ijerph16152800

**Published:** 2019-08-06

**Authors:** Qian Lu, Xia Zou, Yin Liu, Cheng Gong, Li Ling

**Affiliations:** Department of Medical Statistics, School of Public Health, Sun Yat-sen University, Guangzhou 510080, China

**Keywords:** methadone maintenance treatment, tapering phase, heroin abstinence.

## Abstract

Around half of methadone maintenance treatment (MMT) participants choose the tapering phase, however, the guidelines on tapering differ between countries and only include the tapering rate. Physicians need more evidence to guide clinical practice. We aimed to explore a specific tapering strategy to improve heroin abstinence among MMT participants. We conducted a retrospective study from 2006 to 2017 at nine MMT clinics in Guangdong, China, involving 853 participants with 961 treatment episodes. We performed two-level hierarchical logistic regression models to identify tapering phase characteristics associated with heroin abstinence. Among all treatment episodes, 419 (43.6%) were heroin abstinent. Participants who started tapering after 52 weeks, had a taper start dose of less than 60 mg and a taper ratio of less than 5%/week, while a dose reduction in 75%–89% of the tapering weeks provided the highest odds of heroin abstinence. This study highlights the need for a more gradual taper than current guidelines recommend and strongly suggests the inclusion of other tapering phase characteristics. Those who start the tapering phase later, have a lower dose of methadone, with a more gradual rate of taper, and a dose reduction in 75%–89% of the tapering phase increased the odds of heroin abstinence.

## 1. Introduction

Opioid dependence continues to be a serious public health problem. The estimated number of global opioid-dependent users increased from 16.2 million in 2009 to 19.4 million in 2016 [1]. Methadone Maintenance Treatment (MMT) has proved to be one of the most effective methods for dealing with opioid addiction worldwide, and has served millions of participants since the 1950s [2,3,4].

There has, however, been debate and controversy as to whether methadone should be provided indefinitely or whether participants should taper off after a period of stabilization [5]. Most countries have described a framework for MMT dose adjustment that includes a starting dose, titration phase, maintenance phase, and then the participant can choose to enter a tapering phase or to continue with the maintenance phase [6,7,8,9]. Studies have shown that 41%–57% of participants undergo the tapering phase [6,7,8], the reasons for this are said to be due to the severe side effects, pressure from family members, expectations around full recovery and fear concerning methadone’s harm and/or dependency [9,10,11]. This has led some researchers to propose a dose tapering strategy to hopefully increase complete recovery rates [12,13,14]. Canada, the United States, the United Kingdom, and Myanmar have all adopted tapering strategies [6,10,15,16]. Yet, the guidelines vary from country to country and only include the tapering rate. For example, Canada and the United States recommend a 10% reduction in dose every 1 to 2 weeks [6,10], while Myanmar and the United Kingdom recommend no more than 5 mg/week [15,16].

Studies have indicated guidelines need to include other tapering phase characteristics such as taper start dose and taper start week in an attempt to improve heroin users’ abstinence rates, however, there are several limitations [13,14,17,18]. First, there is a dearth of research on dose tapering strategies for heroin abstinence in developing countries, such as China, Myanmar, and India; most studies have been carried out in developed countries, including the United States, the United Kingdom and Sweden [13,14,17]. Second, though these studies showed that participants with a low dose of methadone, and start taper later may improve heroin abstinence, these studies were small-sampled (48–127 subjects) and short (30 weeks–3 years) [13,14,17]. Third, another 10-year cohort study did find when the dose was decreased in 25%–50% of the tapering phase, it provided the highest possibility of medical safety and participant stability following treatment, but the association between the percentage of weeks when the dose was decreased in the tapering phase and heroin abstinence required further study [18]. Therefore, there is a need for a large sample size and long-term study to provide more detailed evidence including taper start dose, taper start week, taper duration, taper rate and the percentage of weeks when the dose was decreased in the tapering phase for the dose tapering guidelines.

In China, 41% of MMT participants undergo the tapering phase [6,18]. The Chinese MMT program is the largest drug treatment program in the world [19], and by 2016 had served over 493,000 participants [20]. In order to provide reference data for policy makers and health authorities to make detailed guidelines on tapering, we conducted a large, longitudinal retrospective study to comprehensively explore the association between tapering phase characteristics and heroin abstinence. This research (1) calculates the most effective tapering rate to achieve heroin abstinence; and (2) analyzes the relationship between other tapering phase characteristics and heroin abstinence at nine MMT clinics in Guangdong province, China. 

## 2. Materials and Methods

### 2.1. Study Setting

This retrospective study was conducted at nine MMT clinics in Guangdong province, China. Guangdong has the most documented drug users in China (582,600) accounting for 17.9% of all known national users [21]. By the end of 2016, 63 MMT clinics had been established in the province [20]. According to the MMT guidelines in China [22], all participants receive unified MMT including the daily administration of methadone. Since take-home doses are not permitted, participants need to go to the clinic, where physicians give professional care and decide the dose of methadone after consultation with participants. Adjustment of methadone dose is available to all participants at any time during MMT. Included in the Chinese MMT guidelines are strategies to adjust the daily dose from starting dose, titration phase (about 7 days) to maintenance phase (more than a month), but there is no mention of tapering. Once participants drop out, defined as absent from MMT for 14 consecutive days or more, they have to start their dose adjustment from the beginning.

### 2.2. Data Collection 

We collected de-identified electronic medical records stored in the National Unified MMT management system from January 1, 2006 to July 30, 2017. Upon participant’s enrolment in the MMT program, demographic information, drug-use related behaviors and infection status (HIV, HCV) were collected. Demographic information and drug-use related behaviors were self-reported and collected by trained clinic staffs. Participants provided blood samples for HIV and HCV serum antibody testing. HCV antibodies were confirmed by HCV enzyme-linked immunosorbent assay (ELISA, Aibo Biotech Company, Shanghai, China). HIV antibodies were first screened at the MMT clinics, using a colloidal gold method (Aibo Biotech Company, Shanghai, China). Positive samples were further confirmed using western blot (Abbott, MP Biomedicals, LLC, Singapore, Singapore) and two ELISAs (ELISA-1, the 4th generation ELISA, bioMerieux bv, The Netherlands; ELISA-2, Beijing BGI-GBI Biotech Company, Ltd., Beijing, China) at the local Center of Disease Control and Prevention. Urine morphine tests were performed on a random day of each month and the results were collected. Methadone was dispended daily, and the date and dose were also collected.

### 2.3. Definitions

#### 2.3.1. Treatment Episode 

A treatment episode was defined as the period between enrolment or re-enrolment in MMT (the first day of the treatment episode) and the following drop out date (the last day of the treatment episode). One participant may have more than one treatment episode.

#### 2.3.2. Tapering Phase Characteristics

Similar to another study [18], the mean daily dose per week (mean weekly dose) for every treatment episode was calculated using: dose_t_ = (∑dose dispensed in week t)/(∑days receiving methadone in week t). A set of variables were constructed to describe the tapering phase: 

(1) taper start week: the first instance after the 6-week point (the shortest length of period from starting dose, titration phase (about 7 days) to maintenance phase (more than a month)), when the mean weekly dose was decreased and then remained constant or decreased further for at least 4 weeks (dose_t-i_ > dose_t_ and dose_t_ ≥ dose_t+i_, i = (1,2,3,4), t ≥ 6) (Figure 1a);

(2) taper start dose: the mean weekly dose of taper start week (Figure 1a); 

(3) taper duration: the period between the taper start week and the final week of the tapering phase (calculated by week) (Figure 1b);

(4) taper dose: the median of dose changes per week when the dose was decreased throughout the tapering phase: the median of |dose_t_ − dose_t-1_|, t = taper start week, …, the final week of the tapering phase, when dose_t_ < dose_t-1_ (Figure 1b);

(5) taper ratio: the median ratio of dose changes per week when the dose was decreased throughout the tapering phase: the median of ((|dose_t_ − dose_t-1_|/dose_t-1_) × 100%), t = taper start week, …, the final week of the tapering phase, when dose_t_ < dose_t-1_ (Figure 1b);

(6) the percentage of weeks when the dose was decreased throughout the tapering phase: the length of weeks when the dose was decreased/taper duration (Figure 1c).

### 2.4. Study Subjects 

Treatment episodes were selected if they: (1) included a tapering phase; and (2) had urine morphine test records available for the tapering phase. A previous study [18] showed that many treatment episodes entered into the tapering phase but reverted to the prior treatment phase, so we selected treatment episodes where the mean weekly dose was either decreasing during the final 4 weeks of the treatment episode or had decreased to no more than 20 mg/day during the final 4 weeks (dose_T-i_ > dose_T_ or dose_T-i_ ≤ 20 mg (i = 0,1,2,3,4 and T is the last week of the treatment episode)).

### 2.5. Measures

#### 2.5.1. Dependent Variable

The definition of heroin abstinence during the tapering phase was based on the results of the urine morphine tests. Participants who did not have any positive urine morphine test results during the tapering phase belonged to the “urine test negative” group, and others belonged to the “urine test positive” group. 

#### 2.5.2. Independent Variables

Tapering phase characteristics: (1) taper start week; (2) taper start dose; (3) taper duration; (4) taper dose; (5) taper ratio; (6) the percentage of weeks when the dose was decreased throughout the tapering phase.

Covariates were added to control potential confounders: (1) characteristics of the whole treatment episode referred to the duration of the treatment episode (calculated by year) and the median rate of dose change throughout the treatment episode: the median of (|dose_t_ − dose_t-1_|/dose_t-1_) × 100%), t = the first week of the treatment episode,…, the final week of the treatment episode; (2) demographic characteristics: gender, age, marital status, education, employment; (3) drug use behaviors: age of initial drug use, intravenous drug use before enrolment in MMT, number of years of drug abuse before enrolment in MMT; (4) infection status: HCV, HIV; and (5) adherence to MMT: treatment attendance, the number of days attending the clinic (taking methadone)/the number of days of the treatment episode. 

The taper dose was classified as <5 mg/week (guidelines’ recommendations) [15,16], 5–10 mg/week and >10 mg/week. The taper ratio was classified as <5%/week, 5%–10%/week (guidelines’ recommendations) [6,10] and >10%/week. Other continuous variables were categorized due to observed distributions, previous studies, and the intention of maximizing clinical interpretability.

### 2.6. Statistical Analysis

All data were analyzed using SAS, version 9.4 (SAS Institute Inc., Cary, NC, USA). We used mean and standard deviation (SD) to describe normally distributed continuous variables, and the median and inter-quartile range (IQR) to describe non-normally distributed continuous variables. Categorical variables were described by frequency distributions and percentages. Since a hierarchical structure resulted from the clustering of treatment episodes (level 1) among different participants (level 2), we used two-level hierarchical logistic regression models to explore the factors associated with heroin abstinence during the tapering phase. All variables with a *p* < 0.20 in univariable regression and all tapering phase characteristics were included in the subsequent multivariable regression. Due to the collinearity and conceptual overlap between the taper dose and taper ratio, we produced two separate multivariable regression models. The goodness-of-fit of these two models were compared using Akaike’s information criteria (AIC). We conducted subgroup analyzes for subsets of participants according to the population characteristics with a *p* < 0.20 in univariable regression. All reported *p*-values were two-sided and considered significant at *p* < 0.05.

### 2.7. Ethical Statement 

This research was approved by Sun Yat-sen University’s School of Public Health’s Institutional Review Board (No: 2013-26).

## 3. Results

### 3.1. Demographic Characteristics, Drug Use Behaviors and HIV/HCV Infection Status 

A total of 853 participants with 961 treatment episodes were included in this study. Each participant had 1 to 4 (median: 1, IQR: (1,1)) treatment episodes. The mean age of the participants was 37.2 (SD = 6.72) years old, the mean age of initial drug use was 23.8 (SD = 6.05) years old, and most of them had abused drug for more than 10 years (78.4%). Participants were predominantly male (90.6%), unmarried (57.0%), educated to middle school level (65.5%), unemployed (76.1%) and had used drug intravenously before enrolment in MMT (89.6%). The HIV and HCV prevalence among participants at MMT enrolment was 7.0% and 79.2%, respectively (Table 1).

### 3.2. Characteristics of Treatment Episode Which Contained a Tapering Phase

In our study, 419 (43.6%) out of 961 individuals were heroin abstinent (urine test negative) during their tapering phase, and 542 (56.4%) individuals were “urine test positive”. Most individuals (72.8%) had a treatment attendance of over 80% and a median rate of dose change throughout the treatment episode of less than 1% (81.0%). The mean duration of each treatment episode was 2.3 (SD = 1.72) years (41.7%) (Table 2).

Only 38.7% of treatment episodes had a taper ratio that was recommended by other countries’ guidelines (5%–10%/week). Most treatment episodes had a faster taper ratio (>10%/week, 50.4%) and most treatment episodes had a taper dose recommended by other countries (<5 mg/week, 77.1%). For 51.4% of treatment episodes, the taper start week was earlier than the 16th week since the treatment episode was initiated. For 65.9% of treatment episodes, the taper start dose was less than 60 mg/day. Most treatment episodes included a tapering phase that was more than 52 weeks (44.3%) with a dose reduction in 25%–49% of the weeks (43.1%). All tapering phase characteristics, except for the taper start week, were statistically significant between different subgroups in the univariable regression analyzes, with a *p*-value < 0.05 (Table 2). 

### 3.3. Association between Tapering Phase Characteristics and Heroin Abstinence

All tapering phase characteristics were associated with heroin abstinence. Among the two models (Table 3), model l which contained the taper ratio provided the lowest AIC (3714.03 vs. 3722.27), indicating the greatest model fit and highest explanatory power. The results of the model which contained the taper dose (model 2) were consistent with the model which contained the taper ratio (model 1).

As shown in model 1, participants who started tapering after 52 weeks (>52 weeks vs. <16 weeks: OR = 2.81, 95% CI: 1.48–5.34), had a taper start dose of less than 60 mg (<60 mg vs. 60–120 mg: OR = 2.08, 95% CI: 1.44–3.00), a taper ratio less than the guidelines’ recommendations (<5%/week vs. 5%–10%/week: OR = 2.08, 95% CI: 1.18–3.64) and the dose was reduced in 75%–89% of the tapering weeks (75%–89% vs. <25%: OR = 3.07, 95% CI: 1.22–7.68) were more likely to be heroin abstinent during the tapering phase. 

Individuals who had a treatment attendance of more than 80% (>80% vs. <50%: OR = 3.40, 95% CI: 1.44–7.98) increased the odds of heroin abstinence. 

### 3.4. Subgroup Analyzes

The effects of all tapering phase characteristics, except the taper ratio, were consistent between unemployed participants and the whole population. For employed participants, only those who had a taper start dose of less than 60 mg (<60 mg vs. 60–120 mg: OR = 1.79, 95% CI: 1.09–2.95) were more likely to be heroin abstinent during the tapering phase (Table 4).

The effects of all tapering phase characteristics, except the percentage of weeks when the dose was decreased throughout the tapering phase, were almost consistent between participants who had abused drug before enrolment in MMT for ≥10 years and the whole population. For participants who had abused drug before enrolment in MMT for <10 years, only who had a taper start dose of less than 60 mg (<60 mg vs. 60–120 mg: OR = 2.87, 95% CI: 1.16–7.11) were more likely to be heroin abstinent during the tapering phase (Table 5). 

## 4. Discussion

Existing guidelines on tapering vary from country to country and only include the taper rate. Studies have indicated that guidelines should also include other tapering phase characteristics such as start week and start dose in an attempt to improve heroin users’ abstinence rates [13,14,17,18]. Our retrospective study provides strong supplementary evidence for their inclusion into current guidelines, we advise participants who want to taper to start the tapering phase after 52 weeks, with a more gradual taper ratio of less than 5%/week and a dose reduction in 75%–89% of the tapering weeks, as this provided the highest odds of heroin abstinence.

Since there are no specific guidelines on tapering in China, when compared with the guidelines recommended by other countries, most of the participants in our research tapered at a faster rate (>10%/week vs. 5%–10%/week). These aggressive tapers underline the urgent need to give more detailed guidelines. In these aggressive cases, outcomes are substantially poorer; participants have higher rates of concurrent heroin use and more withdrawal symptoms [18]. A more gradual taper ratio (<5%/week) may increase the possibility of heroin abstinence, this finding is in agreement with a number of studies which advocate a slow taper [13,14,23] as larger dose decrements are associated with increased dropout rates, illicit narcotics use, and participant distress [13]. We advise a ratio of <5%/week, which is slower than current guidelines recommended by Canada and the United States (10%/1–2 week) [6,10]. In China, the mean dose of methadone is lower than in Canada and the United States [3,7], thus tapering may need to be even more gradual when compared with higher doses to help alleviate symptoms [14]. For countries that prescribe doses as low as China, like Myanmar [24] and India [25], an even more gradual taper rate may be the most suitable.

One standout finding from our study is that reducing the dose in 75%–89% of the weeks is the most effective for heroin abstinence. Participants do not need to taper for the entirety of the tapering phase; which may help with stabilizing and alleviating participant’s withdrawal symptoms [14]. Since the percentage of weeks when the dose was decreased throughout the tapering phase is also associated with medical safety and participant stability following treatment [18], physicians should take this into consideration in future practice.

Our findings, consistent with previous studies, suggest participants should start the tapering phase after a year or more of MMT, as they would be more likely to be heroin abstinent. Longer retention has proved to be associated with better treatment outcomes, such as reducing the risk of relapse, providing a higher level of emotional stability and social rehabilitation [26,27,28]. 

Concurrent heroin use is common among MMT participants, especially among those who begin to taper [6]. The overall rate of concurrent heroin use of participants in the tapering phase was 56.40% in our study, which is similar to previous studies (50.00–72.00%) [12,17,29]. The high risk of concurrent heroin use needs to be explained to participants before initiating the tapering phase and treatment providers should be cautious about encouraging participants to withdraw from the maintenance phase of MMT [30]. 

Our study also indicated that participants with a taper start dose less than 60 mg were more likely to be heroin abstinent than those with a dose of 60–120 mg. Since methadone is used to reduce the severity of the addiction, participants who require higher doses have a higher dependency [31]. Higher prior drug dependency increases the participant’s withdrawal symptoms when decreasing the dose of methadone, thus impacting the possibility of heroin abstinence [32,33]. In addition, the World Health Organization (WHO) (2009) claims treatment attendance is key to MMT success [34]. Our findings support this claim, as high treatment attendance in our study was associated with heroin abstinence. We advise physicians to pay close attention to participants with high doses or low treatment attendance when they try to taper, physicians also need to assess the participant’s motivations when decreasing their dose and to explain the risks.

Variations may exist with respect to the heterogeneity of the population, we did subgroup analyzes for populations of different employment status and years of drug abuse before enrolment in MMT to explore them. We found the effects of tapering phase characteristics were almost consistent between unemployed participants and the whole population. However, for employed participants, only taper start dose was found to be associated with heroin abstinence. In addition to providing income, employment adds important social supports from families and colleagues to MMT participants that discourage continued drug use [35,36]. We also found the effects of tapering phase characteristics were almost consistent between participants who had abused drug before enrolment in MMT for ≥10 years and the whole population. Meanwhile, for other participants, only participants with a taper start dose of less than 60 mg were more likely to be heroin abstinent. Participants who have a shorter time of drug abuse before MMT is associated with a lower dependency of the drug, thus increasing the possibility of heroin abstinence [37]. As the majority of participants had abused drugs for more than 10 years or were unemployed [35,36,37], tapering phase characteristics still play an important role in heroin abstinence. 

The study has several limitations. First, physicians’ decisions about the phases of the treatment episode were not observed which increases the difficulty of classifying the phases. Similar to a previous study [18], we classified the initiation of the tapering phase according to the dose changes, this needs to be considered carefully in future research. Second, several residual or unmeasured confounding variables may be unaccounted for. We were unable to observe addiction severity and the availability of psychosocial care. Finally, this research was a retrospective study, causal relationships between characteristics and heroin abstinence may not be firmly established. 

## 5. Conclusions

In our analysis of nine MMT clinics in Guangdong, China, concurrent heroin use among MMT participants during a tapering phase was common. Our study highlights the need for a more gradual taper than current guidelines recommend and strongly suggests the inclusion of other tapering phase characteristics. Those who start the tapering phase later have a lower dose of methadone, with a more gradual rate of taper, and a dose reduction in 75%–89% of the tapering phase increased the odds of heroin abstinence.

## Figures and Tables

**Figure 1 ijerph-16-02800-f001:**
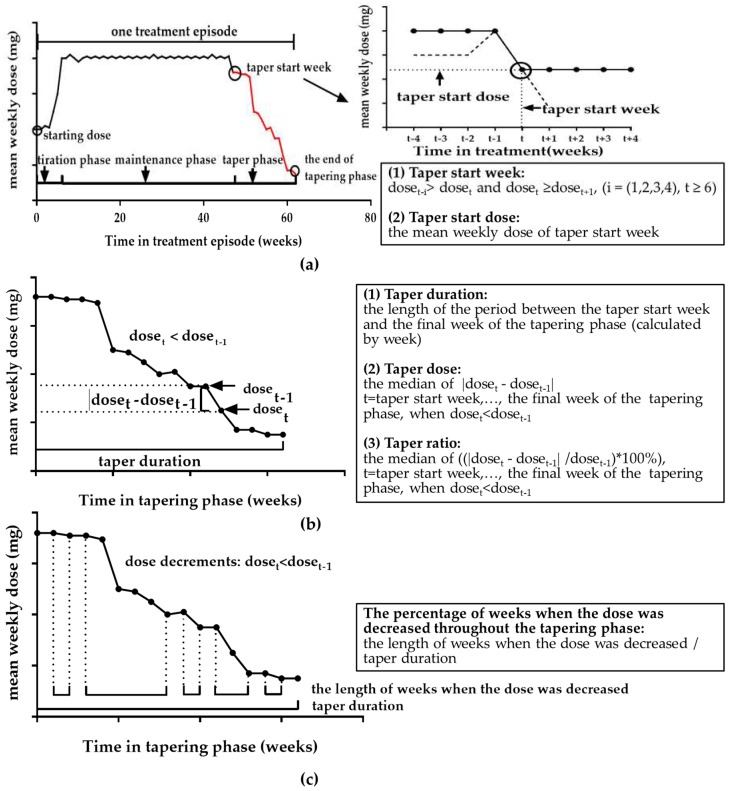
Definition of tapering phase characteristics: (**a**) taper start week and taper start dose; (**b**) taper duration, taper dose and taper ratio; (**c**) the percentage of weeks when the dose was decreased throughout the tapering phase.

**Table 1 ijerph-16-02800-t001:** Characteristics of MMT (methadone maintenance treatment) participants and univariable regression results of heroin abstinence (level 2, *N* = 853).

Variables	No. (%)	cOR (95% CI)
*Demographic characteristics*		
Gender		
male	773 (90.6)	ref.
female	80 (9.4)	0.95 (0.61–1.48)
Age, mean ± SD	37.2 ± 6.72	1.00 (0.98–1.02)
Marital status		
married	367 (43.0)	ref.
single/divorced/widowed	486 (57.0)	0.88 (0.68–1.15)
Education		
primary school or below	138 (16.2)	ref.
middle school	559 (65.5)	1.03 (0.72–1.48)
high school or above	156 (18.3)	0.77 (0.49–1.20)
Employment status		
unemployed	649 (76.1)	ref.
employed	204 (23.9)	1.24 (0.91–1.68) *
*Drug use behaviors*		
Age at initial drug use, mean ± SD	23.8 ± 6.05	1.00 (0.97–1.02)
Intravenous drug use before enrolment in MMT		
no	89 (10.4)	ref.
yes	764 (89.6)	0.86 (0.56–1.30)
Years of drug abuse before enrolment in MMT		
<5	53 (6.2)	1.37 (0.75–2.51)
5–9	131 (15.4)	1.49 (1.01–2.20) **
≥10	669 (78.4)	ref.
*Infection status*		
HIV-infected		
no	781 (91.6)	ref.
yes	60 (7.0)	0.66 (0.39–1.14)
missing information	12 (1.4)	-
HCV-infected		
no	160 (18.8)	ref.
yes	676 (79.2)	0.86 (0.62*–*1.19)
missing information	17 (2.0)	-
No. of treatment episode, median (IQR)	1 (1,1)	

Dependent variable: heroin abstinence during the tapering phase. Independent variables: gender, age, marital status, education, employment status, age at initial drug use, intravenous drug use before enrolment in MMT, years of drug abuse before enrolment in MMT, HIV-infected, HCV-infected. CI, confidence interval. cOR, crude odds ratio. * *p* < 0.20, ** *p* < 0.05.

**Table 2 ijerph-16-02800-t002:** Characteristics of each treatment episode which contained a tapering phase and univariable regression results of heroin abstinence (level 1, *N* = 961).

Variables	Urine Test Negative (*N* = 419)	Urine Test Positive (*N* = 542)	Total(*N* = 961)	cOR (95% CI)
	No. (%)	No. (%)	No. (%)	
*Adherence to MMT*				
Treatment attendance (%)				
<50	12 (2.9)	27 (5.0)	39 (4.1)	ref.
50–80	87 (20.8)	135 (24.9)	222 (23.1)	1.45 (0.69–3.05)
>80	320 (76.4)	380 (70.1)	700 (72.8)	1.90 (0.94–3.85) *
*Characteristics of the whole treatment episode*				
Duration of treatment episode (years), mean ± SD	1.8 ± 1.13	2.8 ± 1.97	2.3 ± 1.72	0.65 (0.59–0.72) ***
% change in dose throughout the treatment episode per week (%)				
<1	320 (76.4)	459 (84.7)	779 (81.0)	ref.
1–4	40 (9.5)	49 (9.0)	89 (9.3)	1.18 (0.75–1.84)
>4	59 (14.1)	34 (6.3)	93 (9.7)	2.51 (1.59–3.95) ***
*Tapering phase characteristics*				
Taper start week				
<16	215 (51.3)	279 (51.5)	494 (51.4)	ref.
16–52	161 (38.4)	218 (40.2)	379 (39.4)	0.96 (0.73–1.26)
>52	43 (10.3)	45 (8.3)	88 (9.2)	1.24 (0.78–1.97)
Taper start dose (mg)				
<60	304 (72.6)	329 (60.7)	633 (65.9)	1.76 (1.32–2.34) ***
60–120	108 (25.8)	206 (38.0)	314 (32.7)	ref.
>120	7 (1.7)	7 (1.3)	14 (1.5)	1.90 (0.64–5.64)
Taper duration (weeks)				
<13	109 (26.0)	54 (10.0)	163 (17.0)	5.24 (3.53–7.80) ***
13–25	97 (23.2)	75 (13.8)	172 (17.9)	3.35 (2.30–4.88) ***
26–52	94 (22.4)	106 (19.6)	200 (20.8)	2.31 (1.62–3.30) ***
>52	119 (28.4)	307 (56.6)	426 (44.3)	ref.
Taper dose (mg/week)				
<5 (guidelines’ recommendations)	343 (81.9)	398 (73.4)	741 (77.1)	ref.
5–10	72 (17.2)	128 (23.6)	200 (20.8)	0.65 (0.47–0.91) **
>10	4 (1.0)	16 (3.0)	20 (2.1)	0.29 (0.09–0.88) **
Taper ratio (%/week)				
<5	52 (12.4)	53 (9.8)	105 (10.9)	1.59 (1.02–2.47) **
5–10 (guidelines’ recommendations)	142 (33.9)	230 (42.4)	372 (38.7)	ref.
>10	225 (53.7)	259 (47.8)	484 (50.4)	1.40 (1.06–1.85) **
% of weeks when the dose was decreased in the tapering phase (%)				
<25	74 (17.7)	137 (25.3)	211 (22.0)	ref.
25–49	147 (35.1)	267 (49.3)	414 (43.1)	1.02 (0.72–1.46)
50–74	112 (26.7)	109 (20.1)	221 (23.0)	1.91 (1.28–2.84) ***
75–89	49 (11.7)	17 (3.1)	66 (6.9)	5.42 (2.88–10.20) ***
≥90	37 (8.8)	12 (2.2)	49 (5.1)	5.82 (2.82–11.99) ***

Dependent variable: heroin abstinence during the tapering phase. Independent variables: treatment attendance, duration of treatment episode, the median rate of dose change throughout the treatment episode, taper start week, taper start dose, taper duration, taper dose, taper ratio, the percentage of weeks when the dose was decreased in the tapering phase. CI, confidence interval. cOR, crude odds ratio. * *p* < 0.20, ** *p* < 0.05, *** *p* < 0.01.

**Table 3 ijerph-16-02800-t003:** Multivariable two-level logistic regression results: predictors of heroin abstinence.

Variables	Model 1		Model 2	
	aOR (95% CI)	*p*	aOR (95% CI)	*p*
Employment Status				
unemployed	ref.		ref.	
employed	0.88 (0.60–1.28)	0.502	0.88 (0.61–1.29)	0.525
Years of drug abuse before enrolment in MMT				
<5	1.43 (0.70–2.90)	0.323	1.52 (0.75–3.09)	0.243
5–9	1.43 (0.92–2.23)	0.117	1.40 (0.90–2.18)	0.139
≥10	ref.		ref.	
Treatment attendance (%)				
<50	ref.		ref.	
50–80	1.72 (0.72–4.11)	0.229	1.62 (0.67–3.92)	0.296
>80	3.40 (1.44–7.98)	0.009	3.29 (1.38–7.87)	0.012
Duration of the treatment episode (years)	0.63 (0.52–0.77)	<0.001	0.64 (0.53–0.78)	<0.001
% change in dose throughout the treatment episode per week (%)				
<1	ref.		ref.	
1–4	0.77 (0.42–1.39)	0.388	0.79 (0.43–1.45)	0.453
>4	1.08 (0.53–2.17)	0.840	0.93 (0.46–1.88)	0.841
Taper start week				
<16	ref.		ref.	
16–52	1.29 (0.90–1.83)	0.170	1.33 (0.93–1.90)	0.122
>52	2.81 (1.48–5.34)	0.003	2.86 (1.50–5.47)	0.003
Taper start dose (mg)				
<60	2.08 (1.44–3.00)	<0.001	1.60 (1.11–2.31)	0.018
60–120	ref.		ref.	
>120	0.87 (0.23–3.26)	0.835	1.36 (0.38–4.87)	0.642
Taper duration (weeks)				
<13	1.77 (0.90–3.50)	0.106	2.15 (0.87–5.34)	0.112
13–25	1.42 (0.78–2.58)	0.255	1.58 (0.86–2.90)	0.142
26–52	1.14 (0.68–1.91)	0.625	1.20 (0.71–2.01)	0.494
>52	ref.		ref.	
Taper ratio (%/week)				
<5	2.08 (1.18–3.64)	0.015		
5–10 (guidelines recommendations)	ref.			
>10	0.82 (0.57–1.18)	0.291		
Taper dose (mg/week)				
<5 (guidelines’ recommendations)			ref.	
5–10			0.55 (0.36–0.84)	0.013
>10			0.18 (0.05–0.62)	0.014
% weeks when dose decreased in tapering phase (%)				
<25	ref.		ref.	
25–50	0.97 (0.63–1.47)	0.872	0.91 (0.60–1.39)	0.669
50–74	1.23 (0.72–2.12)	0.449	1.08 (0.63–1.88)	0.776
75–89	3.07 (1.22–7.68)	0.020	2.71 (1.07–6.85)	0.039
≥90	2.53 (0.93–6.88)	0.075	2.14 (0.78–5.89)	0.145
AIC	3714.03		3722.27	

Dependent variable: heroin abstinence during the tapering phase. Independent variables (model 1): employment status, years of drug abuse before enrolment in MMT, treatment attendance, duration of treatment episode, the median rate of dose change throughout the treatment episode, taper start week, taper start dose, taper duration, **taper ratio**, the percentage of weeks when the dose was decreased in the tapering phase. Independent variables (model 2): employment status, years of drug abuse before enrolment in MMT, treatment attendance, duration of treatment episode, the median rate of dose change throughout the treatment episode, taper start week, taper start dose, taper duration, **taper dose**, the percentage of weeks when the dose was decreased in the tapering phase. CI represents the confidence interval. aOR represents the adjusted odds ratio.

**Table 4 ijerph-16-02800-t004:** Subgroup analyzes of the effect of variables on heroin abstinence in different employment status.

Variables	Employed (*N* = 204)		Unemployed (*N* = 649)	
	aOR (95% CI)	*p*	aOR (95% CI)	*p*
Taper start week				
<16	ref.		ref.	
16–52	0.70 (0.33–1.48)	0.349	1.53 (1.05–2.23)	0.027
>52	2.68 (0.70–10.29)	0.149	2.94 (1.49–5.81)	0.002
Taper start dose (mg)				
<60	1.79 (1.09–2.95)	0.043	1.98 (1.33–2.93)	0.001
60–120	ref.		ref.	
>120	1.14 (0.27–4.79)	0.858	1.10 (0.29–4.11)	0.889
Taper duration (weeks)				
<13	2.48 (0.55–11.25)	0.238	2.03 (1.01–4.10)	0.048
13–25	1.23 (0.33–4.52)	0.760	1.67 (0.90–3.09)	0.106
26–52	0.74 (0.24–2.27)	0.599	1.38 (0.80–2.36)	0.243
>52	ref.		ref.	
Taper ratio (%/week)				
<5	1.83 (0.65–5.16)	0.248	1.72 (0.92–3.24)	0.091
5–10 (guidelines recommendations)	ref.			
>10	1.02 (0.47–2.19)	0.966	0.86 (0.59–1.25)	0.424
% weeks when dose decreased in tapering phase (%)				
<25	ref.		ref.	
25–50	0.61 (0.25–1.52)	0.287	0.85 (0.54–1.34)	0.494
50–74	0.65 (0.19–2.17)	0.479	1.27 (0.71–2.28)	0.427
75–89	1.03 (0.18–5.75)	0.974	2.80 (1.07–7.31)	0.036
≥90	0.91 (0.11–7.26)	0.925	2.90 (0.96–8.79)	0.059

Dependent variable: heroin abstinence during the tapering phase. Independent variables: employment status, years of drug abuse before enrolment in MMT, treatment attendance, duration of treatment episode, the median rate of dose change throughout the treatment episode, taper start week, taper start dose, taper duration, taper ratio, the percentage of weeks when the dose was decreased in the tapering phase. CI represents the confidence interval. aOR represents the adjusted odds ratio.

**Table 5 ijerph-16-02800-t005:** Subgroup analyzes of the effect of variables on heroin abstinence in different years of drug abuse before enrolment in MMT.

Variables	<10 years (*N* = 184)		≥10 years (*N* = 669)	
	aOR (95% CI)	*p*	aOR (95% CI)	*p*
Taper start week				
<16	ref.		ref.	
16–52	0.86 (0.36–2.06)	0.738	1.39 (0.93–2.07)	0.110
>52	4.70 (0.98–22.49)	0.052	2.45 (1.18–5.11)	0.016
Taper start dose (mg)				
<60	2.87 (1.16–7.11)	0.023	1.97 (1.30–3.00)	0.001
60–120	ref.		ref.	
>120	1.49 (0.09–24.14)	0.775	0.69 (0.15–3.23)	0.635
Taper duration (weeks)				
<13	0.58 (0.10–3.40)	0.540	2.45 (1.14–5.25)	0.022
13–25	0.53 (0.12–2.37)	0.407	1.82 (0.92–3.61)	0.085
26–52	0.40 (0.11–1.46)	0.164	1.49 (0.83–2.68)	0.176
>52	ref.		ref.	
Taper ratio (%/week)				
<5	2.05 (0.59–7.07)	0.256	2.26 (1.18–4.32)	0.014
5–10 (guidelines recommendations)	ref.		ref.	
>10	0.73 (0.31–1.72)	0.473	0.85 (0.56–1.29)	0.443
% weeks when dose decreased in tapering phase (%)				
<25	ref.		ref.	
25–50	0.94 (0.34–2.61)	0.897	0.97 (0.60–1.56)	0.894
50–74	1.82 (0.43–7.63)	0.409	1.12 (0.61–2.06)	0.716
75–89	5.80 (0.68–49.40)	0.107	2.51 (0.87–7.23)	0.089
≥90	11.58 (0.62–215.08)	0.100	1.81 (0.59–5.50)	0.297

Dependent variable: heroin abstinence during the tapering phase. Independent variables: employment status, treatment attendance, duration of treatment episode, the median rate of dose change throughout the treatment episode, taper start week, taper start dose, taper duration, taper ratio, the percentage of weeks when the dose was decreased in the tapering phase. CI represents the confidence interval. aOR represents the adjusted odds ratio.

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
