# Peer review of "Dose Tapering Strategy for Heroin Abstinence among Methadone Maintenance Treatment Participants: Evidence from A Retrospective Study in Guangdong, China"

_ijerph, 2019, doi:10.3390/ijerph16152800_

Round 1

Reviewer 1 Report

In the present article, a very important topic is touched. Dose Tapering Strategy for Heroin Abstinence among Methadone Maintenance Treatment Participants is of particular interest in many parts of the world, and this study will be a well read and cited article for similar studies. 

Author Response

July 26, 2019

Dear Reviewer,

We greatly appreciate your comments and appreciate the opportunity to revise and resubmit. Below is the point-by-point reply to your comments.

Point 1: In the present article, a very important topic is touched. Dose Tapering Strategy for Heroin Abstinence among Methadone Maintenance Treatment Participants is of particular interest in many parts of the world, and this study will be a well read and cited article for similar studies. 

Response 1: Thanks for your positive comments.

Yours sincerely,

Li Ling and colleagues

Li Ling,

Professor (Doctoral supervisor), PhD,

Department of Medical Statistics, School of Public Health, Sun Yat-sen University

Fax: 86-20-87335524

Reviewer 2 Report

The manuscript is thorough and well written. All the methods/inclusions followed were well described. Tables and graphs were clearly presented. Conclusions drawn from the results were logical. I would recommend publishing the article after minor grammatical/English corrections. 

Author Response

July 26, 2019

Dear Reviewer,

We greatly appreciate your thoughtful comments that helped improve our manuscript. We have now addressed all of your comments in the revised manuscript. Below is the point-by-point response to your comments.

Point 1: The manuscript is thorough and well written. All the methods/inclusions followed were well described. Tables and graphs were clearly presented. Conclusions drawn from the results were logical. I would recommend publishing the article after minor grammatical/English corrections. 

Response 1: Thanks for your positive comments. We have carefully done English grammar checking in the revised version of the manuscript.

Yours sincerely,

Li Ling and colleagues

Li Ling,

Professor (Doctoral supervisor), PhD,

Department of Medical Statistics, School of Public Health, Sun Yat-sen University

Fax: 86-20-87335524

Reviewer 3 Report

Authors need to modify tables. Table 2 should be eliminated, and tables 3 and 4 should be merged into models 1 and 2. This allows readers to compare the differences between models 1 and 2 in a table.

Author Response

July 26, 2019

Dear Reviewer,

We greatly appreciate your thoughtful comments that helped improve our manuscript. We have now addressed all of your comments in the revised manuscript. Below is the point-by-point response to your comments.

Point 1: Authors need to modify tables. Table 2 should be eliminated.

Response 1: We thank the reviewer for this comment.

There is a hierarchical structure resulted from the clustering of treatment episodes (level 1) among different participants (level 2). We think using two tables to describe characteristics of MMT participants (level 2) and characteristics of treatment episodes (level 1) separately would be more clear and easier to read.

Point 2: Tables 3 and 4 should be merged into models 1 and 2. This allows readers to compare the differences between models 1 and 2 in a table.

Response 2: We thank the reviewer for this good suggestion.

We have merge tables 3 and 4 into models 1 and 2, which can be seen in Table 3.

Yours sincerely,

Li Ling and colleagues

Li Ling,

Professor (Doctoral supervisor), PhD,

Department of Medical Statistics, School of Public Health, Sun Yat-sen University

Fax: 86-20-87335524

Reviewer 4 Report

The manuscript is well written and offers important insight into participants who experience methadone maintenance treatment in China. The study design is robust and the statistical analyses are appropriate. Some major and minor concerns are described below.

Major Concerns:

Some justification should be given for why a crude odds ratio was used in Table 1 instead of an adjusted odds ratio, especially since there was more than one independent variable used in the calculations. The adjusted odds ratio would take into account the effect due to all the additional variables included in the analyses.

The analyses could benefit from understanding how adding weights would impact the data so that they are more closely representative of the overall population.

For Table 1, more information is needed on how the data were collected e.g. for years of intravenous drug use, infection status, education status, etc. Were these self-reported data and if so, how accurate were the patients' responses as a true reflection of their actual experiences?

It would be interesting to explore how individual drug metabolism responses may affect the success of methadone treatment. Variations may exist with respect to the heterogeneity of the population and this aspect should be discussed.

Minor Concerns:

In Table 2, the numbers in the rows are not aligned with the variables which makes reading the table difficult.

Cheng Gong's role on the study is listed as project administration but it is unclear whether or not this contribution merits authorship.

In the tables, the dependent variables and the independent variables should be clearly identified e.g. in footnotes.

Author Response

July 26, 2019

Dear Reviewer,

We greatly appreciate your thoughtful comments that helped improve our manuscript. We have now addressed all of your comments in the revised manuscript. In the enclose response document, we give a point-by-point response to your comments.

Yours sincerely,

Li Ling and colleagues

Li Ling,

Professor (Doctoral supervisor), PhD,

Department of Medical Statistics, School of Public Health, Sun Yat-sen University

Fax: 86-20-87335524

Round 2

Reviewer 4 Report

The revisions to the manuscript are adequate and appropriately answer my comments raised in the initial review.